# Internal tides on the Al-Batinah shelf: evolution, structure and predictability

Gerd A. Bruss<sup>1</sup>, Estel Font<sup>2</sup>, Bastien Y. Queste<sup>2</sup> and Rob A. Hall<sup>3</sup>

<sup>1</sup>Sultan Qaboos University, Muscat, Oman

<sup>5</sup> <sup>2</sup>University of Gothenburg, Gothenburg, Sweden

<sup>3</sup>University of East Anglia, Norwich, United Kingdom

Correspondence to: Gerd A. Bruss (gerd@squ.edu.om)

Abstract. Internal tides are a key mechanism of energy transfer on continental shelves. We present observations of internal tides on the northern Oman shelf based on moored temperature and velocity records collected during summer 2022. The regional shelf exhibits strong summer stratification, supporting shoreward-propagating internal tides with pronounced fortnightly modulation in amplitude and energy fluxes. Despite semidiurnal dominance in barotropic forcing, the internal tides appear predominantly in the diurnal band. Waveform structures undergo transition from quasi-linear depression waves to increasingly nonlinear features, including steepening, asymmetry, and polarity reversal. Modal decomposition shows a shift toward first-mode dominance as the thermocline deepens seasonally. Cross-shelf coherence and phase-speed estimates confirm that the observed internal tides maintain spatial coherence from the shelf edge to the shallow inner shelf beyond the typical internal surf zone. Predictability skill scores indicate that the local internal tides are comparable to high-predictability sites globally while inshore directed energy flux, diurnal dominance and phase lags to barotropic forcing still indicate remote generation.

# 1 Introduction

Internal tides (IT) are internal gravity waves generated by the interaction of barotropic tidal currents with topography in a stratified ocean (Vlasenko et al., 2005). These waves can propagate long distances, transferring energy from generation sites into the open ocean and onto continental shelves (Garrett and Kunze, 2007). As ITs shoal onto continental shelves, they undergo transformation through nonlinear steepening, dispersion, modal scattering, and dissipation (Kelly and Nash, 2010; Lauton et al., 2021). When remotely generated low-mode tides enter the shelf they often evolve into incoherent bores and boluses nearshore, driving turbulence and cross-shelf transport (Walter et al., 2014). Most energy dissipates before reaching shallow depths, with turbulent dissipation matching the flux divergence (Becherer et al., 2021a, b). Shoaling dynamics vary regionally with stratification, slope, and background currents (Masunaga et al., 2024), and energy partitioning among reflection, scattering, and transmission is spatially variable (Siyanbola et al., 2024; Inall et al., 2011). Their propagation is modulated by mesoscale variability and evolving stratification, resulting in both coherent and incoherent components (Kelly et al., 2015; Rayson et al., 2021). While harmonic and response-based models can predict some aspects—especially the coherent fraction—remote forcing and nonlinear interactions contribute to substantial unpredictability (Nash et al., 2012).

The Gulf of Oman (GoO, also known as the Sea of Oman) connects the Arabian Sea and Persian Gulf. The Al-Batinah shelf along northern Oman (between Sohar and Barka) is relatively shallow (~30 m) and wide (~20 km) compared to steeper margins elsewhere. Regional circulation and water mass structure have been described by Johns et al. (2001) and Pous et al. (2004), with later studies expanding on mesoscale eddies (L'Hégaret et al., 2013, 2016), the role of Persian Gulf Water (Queste et al., 2018; Font et al., 2024), and monsoon-driven variability (DiMarco et al., 2023). Seasonal SST patterns and the influence of the summer

monsoon on the Al-Batinah shelf are documented by Al-Hashmi et al. (2019) and DiBattista et al. (2022), while regional warming

and stratification trends have been noted by Bordbar et al. (2024) and Piontkovski and Chiffings (2014). The vertical structure over the slope of the local shelf has been studied using gliders showing a shallow mixed layer above a steep thermocline (TC) in summer, with the TC retreating offshore by November and a new mixed layer reestablishing over winter mode water by April (Font et al., 2022, 2024). Claereboudt (2018) and Chitrakar et al. (2020) provide the few available in situ vertical profiles from the northern

2022, 2024). Claereboudt (2018) and Chitrakar et al. (2020) provide the few available in situ vertical profiles from the northern Omani shelf, showing strong summer stratification and intermittent disturbances. Coles (1997), the earliest local dataset, reported temperature oscillations at sub-diurnal (tidal) frequencies from bottom-mounted thermistors near Fahal Island.

Regional observations of internal waves and tides in the GoO remain sparse. At the Strait of Hormuz, Pous et al. (2004) identified internal wave signatures linked to tidal intensification near the pycnocline. Small and Martin (2002) used SAR imagery and modeling to reveal nonlinear IT packets propagating onto the western shelf. Subeesh et al. (2025) documented seasonally modulated ITs in the western Strait of Hormuz, with enhanced generation during summer. Koohestani et al. (2023) mapped solitary wave hotspots across the basin, linking them among others to barotropic tidal forcing. Despite these advances, the generation, propagation, and impacts of ITs across much of the GoO remain poorly understood, limiting our ability to assess their broader ecological and biogeochemical significance.

Internal tides may have important regional implications. They can enhance vertical mixing and transport low-oxygen water from regional oxygen minimum zones onto the shelf, contributing to seasonal hypoxia, harmful algal blooms, and fish kill events. Additionally, they may provide thermal relief to coral reefs by delivering cooler subsurface waters, thereby protecting corals from heat stress (Storlazzi et al., 2020). Understanding these processes is crucial for predicting ecosystem responses to climate change, managing coastal resources, and improving forecasts of biogeochemical variability linked to internal wave-driven cross-shelf exchange.

Here, we present new observations of internal tides on the Al-Batinah Shelf based on moored velocity and temperature records. The study aims to characterize their temporal variability and vertical structure, quantify energy fluxes and waveform evolution, and assess spatial coherence and predictability in comparison to global observations. These results provide the first detailed view of internal tides in this region and their potential role in local shelf dynamics.

# 60 2 Methods

#### 2.1 Study area

The Al-Batinah shelf lies in the southwest of the GoO in front of the Oman coast between Sohar and Seeb. The average shelf width to the shelf break at 100 m is around 25 km, after which there is a short steep slope followed by more a gradual descend below 250 m down to 3 km (Fig. 1). Within the study area between Al Kaburah and the Sawadi headland, isobaths shallower than 200 m are approximately orientated along the 285-115° axis. The study area lies at a latitude of around 24° with an inertial period of  $\sim$ 29.5 hours (f = 0.81 cpd) and is thus subcritical for the diurnal tidal band. The critical latitudes of the main diurnal harmonics are 30° for  $K_1$  and 27.6° for  $O_1$ . The locations of the two mooring stations at Raqqat As Suwayq (RAS) and Inshore Suwayq (ISQ) are indicated on the map in Fig. 1.

Figure 1: Map of study area and field sampling locations. Bathymetry is from GEBCO Compilation Group (2023). The locations of the on-shelf moorings are shown by yellow markers. Vectors at ISQ represent the energy flux of the diurnal ITs from 15. Aug to 10. Oct. Baroclinic and barotropic flow components are indicated by best-fit ellipses.

#### 2.2 Data and basic diagnostics

On the crest of the small seamount Raqqat As-Suwayq (RAS, Fig. 1), near the shelf edge, temperature at approximately 18 m depth and currents were recorded with an Acoustic Doppler Current Profiler (ADCP) over a 22-month period from Jan 2021 to Oct 2022. Depth average on-shelf stratification was derived by comparing temperatures recorded by near-bottom sensors at RAS with satellite-derived sea surface temperature (SST). To resolve the vertical thermal structure of the oscillations observed in the bottom temperature, a full-depth mooring was deployed at ISQ in 23 m water depth (Fig. 1).

At the ISQ mooring, current profiles were recorded at 5 minute intervals using a bottom-mounted, upward-looking ADCP (AWAC 600 kHz, with 0.5 m bin-size) for approximately four months in winter 2021/22 (Oct–Jan) and three months in summer 2022 (min Jul - mid Oct). Concurrently, during the summer deployment, a time series of the temperature profiles was recorded at the same station using a vertical thermistor chain of HOBO loggers with ~2 m vertical and 15 temporal resolution. High sensor drift in conductivity probes, primarily due to strong biofouling on the shallow shelf, prevented reliable long-term salinity measurements at similar vertical and temporal resolution. Salinity data are therefore limited to discrete CTD casts. For calculations requiring density (e.g., N² or potential energy anomaly), a constant salinity value was used. This value represents the mean salinity from all available CTD casts near the ISQ mooring location during four summer periods between 2021 and 2024 (cf. Fig. 2).

Seawater density  $\rho$  and buoyancy frequency  $N^2 = -g/\rho \cdot (d\rho/dz)$  were calculated using the GSW Oceanographic Toolbox according to TEOS-10 (McDougall and Barker, 2011) and we also calculate vertical shear  $S^2 = (du/dz)^2$  and Richardson number  $Ri = N^2/S^2$ . To define the depth of the mixed layer we use conventional definitions and define thermocline elevation  $z_{TC}$  as the depth at which the temperature  $\theta$  falls below a certain threshold relative to the surface temperature:  $z_{tc}(t)$ :  $\theta(z,t) = \theta_{surf}(t) - \Delta\theta$ .

The value of  $\Delta\theta$  is determined by fitting the variation of  $z_{TC}$  over the record period to match the variation of the vertical level of the upper  $N^2$  peak. For the data from ISQ  $\Delta\theta=2^{\circ}C$  which is larger than commonly used threshold values around 1 °C.

Information on the barotropic tide was derived from the pressure sensor of the ADCP as well as from the TPXO regional tidal model solution for the Persian Gulf / Gulf of Oman (Egbert and Erofeeva, 2002). Satellite derived SST was obtained from the OSTIA dataset (Good et al., 2020).

# 2.3 Background conditions and tidal-band decomposition

To estimate background conditions, except for vertical  $N^2$  profiles, we apply a low-pass filter in the frequency domain with a tapered cutoff at 50 hours and denote background (low-pass filtered) quantities with an overbar. To determine background

stratification ( $N^2$  profiles) we do not average the available temperature profiles over a tidal cycle as this would diffuse  $z_{TC}$  over a wider depth range, thus reducing the  $N^2$  peak. We rather consider background conditions at the times when the instantaneous  $z_{TC}$  crosses the low-pass filtered  $z_{TC}$  (see Sect. 2.4).

To further isolate variability in the diurnal (D) and semidiurnal (SD) frequency bands, we apply wavelet-based decomposition to the baroclinic signals using the maximal overlap discrete wavelet transform (MODWT) with a Symmlet-4 wavelet. Reconstruction within the 8-16 h (SD) and 16-32 h (D) bands is used to extract time-localized tidal variability. Note that the D band also contains energy near the local inertial frequency ( $\sim$ 28 h period).

The barotropic tidal transport from the TPXO model is decomposed as the sum of predicted contributions from tidal constituents in each band:  $D(K_1, O_1, P_1, Q_1)$  and  $SD(M_2, S_2, N_2, K_2)$ . Amplitude spectra are computed using standard FFT-based methods to characterize the stationary (phase-locked) components of the signal, and tidal harmonic analysis is performed using the UTide package (Codiga, 2011).

## 2.4 Properties of Internal Tides

Individual internal tides are defined between upwards crossings of the instantaneous  $z_{TC}$  through the low pass filtered  $z_{TC}$ . Individual tides were identified from the original signal as well as for the decomposed D and SD bands. All individual tides identified by the automatic zero-crossing detection were visually verified. Periods where individual ITs could not be clearly identified were omitted from further analysis and appear as gaps in the respective figures. Variables like the root mean square of the thermocline displacement  $\xi_{rms}$ , phase speed  $c_p$ , wavelength L, energy flux  $F_E$  and wave asymmetry/polarity are determined by integrating over each of the individual ITs.

As ITs shoal on the inner shelf, they steepen and become increasingly nonlinear, ultimately forming sharp bore fronts (Becherer et al., 2021a). Wave asymmetry is quantified by the skewness of the temporal gradient  $d\xi/dt$ , computed within each wave. Positive skewness indicates a steeper leading edge relative to the trailing face. The polarity of an internal wave distinguishes between waves of depression and waves of elevation. Although this concept is more commonly applied to internal solitary waves (ISWs), we also assess a simplified polarity metric defined as the ratio of the maximum upward to downward displacement.

# 2.5 Spatial coherence

To assess the relationship between the recorded baroclinic signals at ISQ and the local surface tide, we analyze the barotropic tidal transport from the TPXO model at the local shelf break. We examine the phase relationship between the fortnightly amplitude modulation of the barotropic transport and that of the internal tides. This analysis is performed in both the D and SD bands by extracting the amplitude envelope using the Hilbert transform of the complex velocity time series (u + iv).

Time lags between signals at RAS and ISQ are determined from three months of temperature records at both stations at 18 m depth. Tidal events within the extended tidal band (6–36 h) are isolated, and time lags are estimated by cross-correlating each ISQ event with the RAS record. Both signals are locally normalized prior to correlation. Phase speeds are then computed by projecting the RAS–ISQ separation onto the direction of the corresponding IT energy flux.

Another simple phase speed estimation of long internal waves in a two-layer, non-rotating, inviscid fluid system is given by the classical shallow-water approximation, which depends on the density contrast between layers, gravitational acceleration, and the harmonic mean of the layer depths:  $c_p = \sqrt{(\Delta \rho/\rho)g\ H/(h_1 + h_2)}$  where  $\Delta \rho$  is the density difference between the two layers,  $\rho$  is a

35 reference density (typically the lower layer), g is the gravitational acceleration, and h<sub>1</sub>, h<sub>2</sub> are the layer thicknesses. Phase speeds determined via these two methods are compared to estimates obtained from modal analysis.

#### 2.6 Internal tide energy flux, PEA and KE

We decompose flow and pressure and determine internal wave energy flux following Kunze et al. (2002) and Kelly et al. (2010). The baroclinic component of the flow  $\mathbf{u'}$  is obtained by  $\mathbf{u'}(z,t) = \mathbf{u}(z,t) - \overline{\mathbf{u}}(z,t) - \mathbf{u}_0(t)$  in which  $\mathbf{u}$  is the instantaneous (measured) current,  $\overline{\mathbf{u}}$  is the low pass filtered flow and  $\mathbf{u}_0$  is determined via the baroclinicity condition  $\frac{1}{H} \int_{-H}^0 \mathbf{u'}(z,t) \, dz = 0$  which requires that  $\mathbf{u'}$  vanishes when integrated over the water column (e.g. Kunze et al., 2002). The removal of the low-frequency component  $\overline{\mathbf{u}}$  isolates the internal wave motions, particularly those at super-inertial frequencies (e.g., D and SD bands). Because the background conditions vary gradually and significantly over the long record, all-time means are unsuitable here. Removal of the low-frequency component is furthermore important since the background flow is also baroclinic.

The energy flux of the internal waves is calculated as  $\mathbf{F_E} = \langle \mathbf{u'p'} \rangle$ . The brackets  $\langle \cdot \rangle$  denote a time average, typically over a moving window or over one phase of the internal tide. The pressure perturbation p' due to the internal waves is obtained by vertical integration of the density anomaly  $p'(z,t) = \int_z^0 \rho'(z',t)g \, dz' + p_{surf}(t)$  The surface pressure  $p_{surf}$  results again from applying the baroclinicity condition (i.e. the pressure perturbation must vanish when integrated over the depth). The density anomaly  $\rho'(z,t)$  is estimated from the vertical displacement of isotherms  $\xi(z,t)$  using the linearized equation of state under the Boussinesq approximation:  $\rho'(z,t) = (\bar{\rho}(z)/g) \, N^2(z) \, \xi(z,t)$ . We use the displacement of isotherms rather than isopycnals due to the lack of high resolution salinity data as described above. Isotherm displacement is calculated as:  $\xi(z,t) = -\theta'(z,t)(\partial z/\partial \bar{\theta})$  where  $\theta'$  and  $\bar{\theta}$  denote baroclinic and low pass filtered temperature respectively.

We use Potential Energy Anomaly (PEA) as a measure for the overall strength of the water column stratification. We use the definition of PEA that was first proposed by Simpson (1981):

$$PEA(t) = \frac{1}{D} \int_{-H}^{\eta} (\bar{\rho}(t) - \rho(z, t)) gz \, dz \tag{1}$$

where D is the total water depth from the sea floor at –H to the free surface  $\eta$  and  $\bar{\rho}$  is the depth averaged density.

PEA quantifies the energy required to fully mix a vertically stratified water column to a state of uniform density. In this study, we focus on the low-frequency variability of PEA and therefore neglect the effects of tidal deformation of the water column, which can be significant when tidal amplitude is comparable to the water column thickness (Hamada and Kim, 2021). We compute PEA at the full temporal resolution of our dataset and subsequently derive daily-averaged values (Fig. 3b). Kinetic energy KE = 1/2  $\rho \left( \overline{(u^2 + v^2)} \right)$  is computed for unfiltered flow, band-pass filtered components, and for barotropic and baroclinic currents. Bulk values are obtained by averaging over depth and over summer and winter periods.

#### 165 2.7 Dynamic modes

We determine vertical dynamic modes by solving the vertical structure eigenvalue problem. The governing equation for vertical displacement or vertical velocity amplitude is derived from the linearized, non-hydrostatic, Boussinesq primitive equations (Cushman-Roisin and Beckers, 2011), resulting in a Sturm–Liouville eigenvalue problem of the form:

$$\frac{d^2}{dz^2}\Phi(z) + \frac{N^2(z)}{c_z^2}\Phi(z) = 0$$
 (2)

which we solve for the boundary conditions  $\Phi(-H) = \Phi(0) = 0$ . To quantify modal energy partitioning, we project the observed baroclinic velocity fields onto the horizontal modes  $p(z) \propto d\Phi(z)/dz$  which are L<sup>2</sup> normalized such that  $\int_{-H}^{0} p^2(z)dz = 1$ . Modal amplitudes are then computed as:

$$A_n(t) = \int_{-H}^0 \mathbf{u'}(z,t) p_n(z,t) dz$$
 (3)

where  $\mathbf{u'}$  is the observed baroclinic cross shelf current. The normalization ensures that  $A_n^2$  is proportional to the energy content in mode n, enabling consistent comparison of relative energy between modes and across time segments. The vertical modes shown in Fig. 4 are scaled with these modal amplitudes. From the eigenspeeds  $c_e$  we determine phase speed under the influence of rotation via the dispersion relation as in Rainville and Pinkel (2006):

$$c_p \equiv \frac{\omega}{(\omega^2 - f^2)^{1/2}} c_e \tag{4}$$

#### 2.8 Predictability

In order to determine the predictability of the internal tides we apply the methods following Nash et al. (2012) (based on the skill score by Murphy, 1988). Predictability is quantified through the analysis of a skill score (SS, Eq. (5)) that measures how much of the variance of the signal is captured by harmonic analysis in moving windows with different window size T. We perform tidal harmonic analysis with the UTide package from Codiga (2011).

$$SS_T = 100\% \times \left[ 1 - \frac{\left( (\Psi - H_T(\Psi))^2 \right)}{\left( \Psi^2 \right)} \right] \tag{5}$$

To enable direct comparison with the global predictability estimates reported by Nash et al. (2012), we applied harmonic fits using the same six tidal constituents (M2, S2, N2, K1, O1, J1), even for short time windows (as short as 5 days). While individual constituents are not spectrally resolvable over such durations, their combined fit offers a consistent measure of variance within the broadened tidal band. A predictability timescale T<sub>50%</sub> is defined as the window length at which the variance explained by the harmonic fit declines to halfway between its maximum (at short durations) and its asymptotic long-term value, approximated by the fit at T = 180 days. This metric reflects the temporal coherence of the internal tide and captures the rate at which phase and amplitude relationships de-correlate under evolving background conditions. While the two summer temperature records at RAS span 180 days, the ISQ records are shorter (Oct 2021–Jan 2022 [winter], and Jul–Oct 2022 [summer]). For these shorter records, we fit a power-law decay of the form SS(T) = a / Tb to estimate the Skill Score at 180 days, acknowledging the associated limitations.

#### 200 3 Results and Discussion

The 22-month temperature record from the shelf-break mooring at RAS (Fig. 2a) shows that during winter (Dec-Mar), the temperature at 18 m depth approximately matches the remotely sensed daily averages of foundation sea surface temperature (SST), indicating fully mixed conditions and small variations in the tidal band. The lowest temperatures, around 23 °C, occur in February. Re-stratification begins in March, marked by a simultaneous increase in both SST and bottom temperature, accompanied by a growing divergence between them. During summer, high surface temperatures exceeding 32 °C appear in early June in both years.

Stratification then breaks down beginning in October, as SST decreases while bottom temperature rises, leading to temperature homogenization by December.

The main differences between the two recorded years stem from inter-annual variability in the local influence of the southwest monsoon over the Arabian Sea between June and September. In 2022, persistent easterly winds from June to October sustained offshore surface transport (not shown), resulting in prolonged low-frequency upwelling. The lowest water temperatures occurred in early August, followed by a gradual warming as the winds weakened. Similar dynamics have been described by DiMarco et al. (2023) for the shelf off Sohar, 150 km to the northwest.

Figure 2: Temperature, salinity, and stratification in the study area. (a) Twenty-two-month temperature record from the RAS mooring at 18 m depth (5-min and daily averages) and satellite-derived daily SST. (b) T-S diagram of all available on-shelf CTD casts from four summer periods (2021–2024) near the ISQ station. Contour lines indicate isopycnals; the green vertical line marks the approximate average salinity used in this study. (c) Vertical profiles of temperature (T), salinity (S), and buoyancy frequency (N²). Solid and dashed black lines represent CTD casts taken 4.5 h apart during the flood phase of the internal tide on 25 August 2022. Thin grey lines in the temperature panel show profiles from the vertical thermistor chain. Green lines indicate constant salinity; the red line shows background stratification.

Shorter, more intense summer upwelling events occasionally reached the surface (e.g., in July 2021), although a vertical temperature gradient was typically retained. Because the in situ temperature record at RAS generally reflects conditions below the shallow thermocline, the temperature difference between surface and bottom serves as an indicator of upper-water-column stratification strength. Between the two years, summer stratification was stronger in 2022, when the daily averaged vertical temperature difference from the surface to 18 m ranged between 4–7 °C, peaking at 9.3 °C on 25 June 2022. Daily variations in bottom temperature at RAS during summer averaged around 4 °C, with maxima up to 8 °C. A spectral analysis of the RAS record is presented in Sect. 3.4.

From CTD casts, we frequently observe a salinity inversion in summer, with higher salinity above z<sub>TC</sub>. Fig. 2b shows data from 47 CTD casts near the ISQ mooring location during four summer periods between 2021 and 2024. The salinity inversion is reflected

in the positive slope of the linear fit in the T–S diagram and is attributed to surface evaporation combined with limited vertical mixing across the thermocline. This inversion reduces the vertical density gradient, thereby weakening the actual pycnocline (N² peak). Consequently, variables derived from stratification—such as IT energy flux or potential energy anomaly (PEA) estimates—may be slightly overestimated if calculated solely from thermal stratification. We assess this effect to be small (see below). Despite the salinity inversion, salinity variability is relatively low compared with temperature, both vertically and spatially across the shelf and over the years surveyed. To calculate density in the absence of high-resolution salinity measurements for the ISQ summer data, we therefore use an average salinity from all CTD casts. The green vertical line in the T–S diagram indicates this average salinity value.

Two CTD casts at the ISQ station on 25 August 2022 (Fig. 2c) illustrate short-term variability during rising internal tides. Offshore bottom flow produced a temporary three-layer structure, with a 6 m uplift of the thermocline and corresponding halocline displacement (Fig. 4). Salinity increased in the lower layer during the flood phase, likely reflecting either intrusion of Persian Gulf Water or subduction and offshore transport of high-salinity inner-shelf water by cascading, down-welling, or internal tides (Shapiro et al., 2003). A shallow diurnal warm layer of 0.8 °C can be seen in the upper 1.4 m of the second profile accompanied by a small rise in surface salinity. Only two salinity profiles are available for this day. We retain the upper halocline gradient as background stratification and approximate the deeper profile from the earlier cast, excluding the bottom anomaly. The resulting N² profiles show that including the observed salinity inversion reduces the stratification peak by <5% relative to constant salinity. This small effect supports the use of temperature-based density estimates for the 11-week energy flux calculations.

#### 3.1 Evolution of Internal Tides

Fig. 3 presents the two main observed variables (temperature and horizontal currents, Fig. 3b-c) from the ISQ mooring, together with derived properties of the internal tides. The dataset spans 11 weeks towards the end of summer, covering both summer conditions and the autumnal transition of stratification. For a short period at the beginning of the record, stratification is weak due to a preceding wind-driven upwelling event (not shown). During the first week of August, stratification is re-established, with a shallow  $z_{TC}$  at approximately 8 m below the surface (15 m above the bottom), and surface temperatures exceeding 32 °C (Fig. 3b). After mid-August, the mean  $z_{TC}$  gradually deepens. During this period, the Potential Energy Anomaly (PEA) remains, on average, around 35 J m<sup>-2</sup>, until stratification eventually breaks down on the shelf (Fig. 3b). The peak and depth-averaged values of the buoyancy frequency are typically  $N^2 \approx 0.01 \text{ s}^{-2}$  and  $N^2 \approx 0.001 \text{ s}^{-2}$ , respectively (Fig. 2c). Similar summer values of  $N^2 \approx 0.001 \text{ s}^{-2}$  have been reported by Font et al. (2022) for the regional shelf slope, and in WOA climatological fields for the central GoO (Koohestani, 2023). Compared to other regions (e.g., compiled in Becherer, 2021b), the Al-Batinah shelf lies at the high end of global stratification intensity (in the absence of river plumes). With shear in the order of  $S^2 \approx 0.0005 \text{ s}^{-2}$  and Richardson numbers mostly  $Ri > \frac{1}{4}$ , conditions across the thermocline remain predominantly stable.

In the tidal band (8–30 h), the cross-shelf baroclinic currents covary in intensity with the displacement of the thermocline (Fig. 3c), which reaches up to 14.5 m towards the end of the record. The recorded ITs displace between 40% (25 August) and 60% (5 Oct) of the water column. As ITs shoal on the sloping shelf, their amplitude eventually approaches the local water depth (Becherer et al., 2021b). The ITs observed at ISQ under typical summer conditions have not yet reached this saturation point.

Figure 3: Mooring data during summer 2022 at station ISQ. (a) Skewness of  $d\xi/dt$  (positive indicates steeper leading edges, negative indicates steeper trailing edges). (b) Sea level (from ADCP pressure sensor), vertical temperature profile, instantaneous  $z_{TC}$  (grey line), and Potential Energy Anomaly (PEA, cyan). (c) Baroclinic cross-shelf currents (coloured) with low-pass filtered  $z_{TC}$  (grey line) and diurnal tidal extrema (grey triangles indicate wave peaks and polarity). (d) RMS of  $\xi$  (bars) and vectors of baroclinic energy flux; lines show amplitude modulation of barotropic tidal transport. Variables in panels (a) and (d) are decomposed into D (grey) and SD (black) components.

Both the root mean square of the thermocline displacement ( $\xi_{rms}$ ) and, even more clearly, the energy flux  $F_E$  reveal the dominance of the D band in the ITs (Fig. 3d). SD ITs can be distinguished in  $\xi_{rms}$ , particularly when their "spring-tide" coincides with periods of low intensity in the D-band.  $F_E$  in the SD band is however largely negligible.  $\xi_{rms}$  and  $F_E$ , in both the D and SD bands, exhibit distinct fortnightly patterns (spring-neap cycles in the SD band), with increasing intensity towards the end of the record. To represent potential local forcing, the amplitude modulation of the barotropic tidal transport at the local shelf break is shown separately for the D and SD bands, each normalized by its standard deviation (lines in Fig. 3d). The beat periods of the main tidal constituents are  $K_1 + O_1$ : 13.66 days, and  $M_2 + S_2$ : 14.77 days. As a result, the two fortnightly peaks shift their relative phase and during the record period, the two cycles are roughly out of phase.

The fortnightly cycles of the ITs are less uniform than the astronomical forcing and exhibit delays of several days relative to the barotropic transport, with these lags decreasing towards the end of the record (Fig. 3d). The SD barotropic transport is, in absolute

terms, approximately 4.5 times stronger than the D-band transport, marking a clear contrast with the ITs, where the D-band dominates. Following full re-stratification after the early August mixing/upwelling event, and from mid-August onwards, the IT energy flux is predominantly directed onshore. According to Siyanbola et al. (2024), this is strong evidence for remote generation. For the shelf off central California, Becherer (2021a) observes that the IT energy flux decreases to near zero by the 25-m isobath. In our data, at 23 m depth,  $F_E$  remains small but distinctly observable.

The observed variability in IT energy flux direction suggests remote generation and propagation effects, rather than purely local forcing. The variability we observe may result from multipath propagation, refraction by spatially and temporally varying stratification or mesoscale currents, and possible interference between remotely generated ITs. Gong et al. (2019) showed that remotely generated ITs can dominate and modulate local energy conversion on a shelf, altering both magnitude and direction of energy flux. Ansong et al. (2017) reported substantial temporal variability in SD IT energy fluxes driven by remote generation and propagation pathways. Therefore, the evolving flux directions in our data likely reflect remote IT sources, which are being refracted and temporally modulated before reaching the ISQ mooring. One potential generation region across the GoO is discussed in Sect.

3.4

At certain periods, decreases in PEA coincide with increases in IT intensity, suggesting episodes of mixing. Nevertheless, Ri across the upper thermocline remains generally high, and the onshore  $F_E$  combined with displacement amplitudes that remain small relative to the water depth (indicating that ITs have not reached the saturation point), suggests that the primary dissipation of IT energy occurs further shoreward. Several studies have reported the transformations long ITs undergo as they shoal on the inner shelf (e.g., Holloway et al., 2003; Colosi et al., 2018). Upon interacting with the seafloor, they develop increasingly steep waveforms, resulting in sharp bore-fronts and/or the generation of higher-frequency wave trains of ISWs/NLIWs. In our data, we observe only the onset of bore-like features towards the end of the record, as IT become increasingly asymmetric (slope skew) and eventually reverse their displacement polarity to waves of elevation in October. While our data does not resolve high frequencies, we observe sporadic variability at periods of approximately 5 min, which could indicate increased ISW/NLIW activity.

Our data suggest that IT at the ISQ location have not yet fully entered their surf zone as defined by Becherer (2021b). Becherer et al. (2021a, b) identify stratification as a primary constraint on the cross-shelf transport of IT energy. When stratification is weak, dissipation occurs farther offshore; whereas under stronger stratification, a greater fraction of energy reaches shallower sites. The strong summer stratification on the Al-Batinah Shelf appears to reduce displacement amplitudes enough and provide a strong enough waveguide to allow IT to propagate farther inshore than in other regions. In linear theory, internal wave energy flux scales with stratification and displacement amplitude as: F<sub>E</sub> ~ ρ<sub>0</sub> N ω η<sup>2</sup>. For our data averaged over the diurnal ITs, log-log regression yields a significant empirical scaling of F<sub>E</sub>αPEA<sup>-1</sup>N ω η<sup>1.7</sup> where PEA is used instead of N as a more robust representation of stratification. This suggests that general conditions remain close to linear. However, towards the end of the record, F<sub>E</sub> increases beyond the general scaling. This increase coincides with a steepening of the leading face of the ITs (positive skewness), suggesting that increasing nonlinearity is the primary driver of the enhanced F<sub>E</sub>.

The increase in nonlinearity coincides with the deepening of  $z_{TC}$  around 19 September, when the low-pass filtered  $z_{TC}$  reaches a level above the local seabed ( $\sim$ 7 m), roughly equivalent to twice the displacement amplitude. After this point, and until the thermocline finally retreats from the shelf (around November; see Font et al. (2024), the ITs at ISQ become increasingly nonlinear. The combination of stratification, bathymetry and  $z_{TC}$  determines where the regional ITs enter their "surf zone".

## 3.2 Waveform Structure

To characterize the waveform structure, we detail the baroclinic currents and isotherms for five periods with pronounced IT activity (Fig. 4). Both fields are band passed over the entire tidal band (D and SD). The wave forms all deviate to varying extent from a

purely sinusoidal shape expected under linear theory. Around 26. Aug when the mixed layer is shallow and the diurnal ITs tides are dominant, the peak isotherm displacement is larger downwards with intensified near bottom currents and the wave form has a steeper leading face (Fig. 4a). Around 3. Sep. when IT are pronounced semidiurnal, the waveform still resembles a depression wave but the slope asymmetry is sometimes reversed, with the steeper gradient now on the trailing face. Despite the clear SD signal in the displacement, the D band is still prominently visible in the baroclinic currents. While the mean  $z_{TC}$  around 10. Sep is slightly lower, the diurnal wave patterns are still similar to the ones in late August. Around 25 Sep the mean  $z_{TC}$  further descended to below 10 m height and the downwards displacement of the ITs extends close to the seafloor, increasing bottom influence on wave structure. The wave trough is flattened and the rising phase is further steepened. Eventually in October the polarity has reversed to higher upwards displacements and a steepening of the falling phase.

Shroyer et al. (2009) describes polarity reversal of NLIW with a shift to elevation waves when the pycnocline, which initially supports depression waves, becomes closer to the bottom than the surface. For shoaling depression waves that undergo polarity reversal, the trailing face can steepen. Similar to our observations, Cai et al. (2012) observe that with the mixed layer depth changing over the seasons, both elevation and depression waves may occur in the same region.

Figure 4: Baroclinic cross-shelf currents in the tidal frequency band (D + SD) for four selected  $\sim$ 4-day intervals. Time of day is aligned across all time-series panels. Colours indicate currents (scale as in Fig. 3); thin grey lines are isotherms, thick grey lines represent instantaneous  $z_{TC}$  and dash-dotted lines show low-pass filtered  $z_{TC}$ . Vertical yellow lines mark one specific tidal phase used to determine the vertical modes. Outside panels show the first two vertical modes, scaled by the energy of currents projected onto pressure modes. Top and bottom panel rows show temperature anomalies of records at RAS and ISQ normalized for each wave segment. Dotted and solid lines indicate original and time shifted signals at RAS respectively.

On the 25 Aug, the phase relationship between the cross shelf currents and the displacement is close to quadrature as would correspond to linear wave dynamics (e.g. Cushman-Roisin and Beckers, 2011). This phase relation is however restricted to the currents within a few meters around  $z_{TC}$ . Currents in lower layers are phase shifted indicating the influence of higher modes. At the end of Sep and the beginning of Oct, isotherm displacement and currents across the whole water column are in phase. This is consistent with the transformation of the ITs into bore like shapes. Duda (2004) assesses phase relation between baroclinic currents and displacement and links deviations from quadrature to increasing nonlinearity during the shoaling of D band ITs. Similar to our observations, Becherer et al. (2021b) describes peaks in near bottom currents as part of steep mode one internal tidal bores.

The dynamic vertical modes shown in the right column of each panel were derived from zero-crossing  $N^2(z)$  profiles, normalized and scaled by the energy of baroclinic currents projected onto each mode. These mode shapes represent the vertical energy

distribution within the tidal band for each period. In late August and early September, energy is distributed between the first and second modes, corresponding to observed vertical shear and phase shifts indicating higher-mode contributions. In contrast, from late September onward, the first mode increasingly dominates, with second-mode contributions diminished. The shift toward dominance of the first mode in late September and October coincides with the observed deepening of the thermocline (cf. Fig. 3), suggesting that stratification weakening favors lower-mode vertical structures. This evolution toward first-mode dominance, waveform steepening, and coherent vertical structure is consistent with increasing nonlinearity during IT shoaling. For instance, Duda and Rainville (2008) observe that as diurnal ITs propagate upslope, they exhibit current minima near the center of the water column, indicating that the energy organizes predominantly into mode-1. They observe nonlinear steepening during upslope propagation, particularly at shallower mooring sites, with saw tooth-like waveforms and bore-like fronts reported as signatures of nonlinear deformation.

The increased mode-1 dominance and waveform steepening we observe in early October align closely with increases in IT amplitude and skewness identified earlier (Fig. 3), highlighting nonlinearity as a primary factor shaping wave evolution. The observed onset of bore-like structures and first-mode intensification may enhance near-bottom mixing, consistent with the transient reductions in PEA (cf. Fig. 3b and 5c).

In the absence of spatial data, we do not apply KdV models to our IT observations. However, others have done so successfully—for example, Holloway et al. (2003) used a modified KdV framework to model IT propagation on the NW Australian shelf. The nonlinear features we observe, such as wave steepening, polarity reversal, and changes in vertical structure, are consistent with established nonlinear internal wave dynamics (e.g., Shroyer et al., 2009; Cai et al., 2012), suggesting that a KdV-type framework could be appropriate for future spatially resolved analyses.

#### 3.3 Cross-shelf Coherence

Cross-shelf coherence measures how consistently IT signals are preserved between offshore and inshore locations, offering insight into their propagation pathways and the degree of energy loss or transformation across the shelf. Fig. 5 shows the temperature recorded at the shelf edge mooring (RAS) together with a temperature time series taken at the same vertical level (18 m depth) from the thermistor chain at the inshore station (ISQ). At the beginning of the period, the influence of the ITs at 18 m is small (though still clearly distinguishable), until the thermocline descends in late August (cf. Fig. 3bc) and the isotherm displacement more prominently reflects in the fixed depth temperatures.

The pattern of high wavelet coherence between the two temperature time series matches with the periods of high IT energy in each of the tidal bands (bars of  $\xi_{rms}$ , Fig. 5a) which indicates that the propagation of the ITs can coherently be tracked from the shelf edge to the 12 km distant mid-shelf location at ISQ until the end of September. This confirms that dissipation and the corresponding discoherencing of the ITs happens inshore of ISQ. It also confirms that the influence of diurnal warm layer dynamics that can reach down to around 6 m on the shallow inner shelf is not interacting significantly with the ITs at ISQ. High coherence generally suggests near linear wave propagation and the dominance of a single propagation direction during the event (not necessarily constant over longer periods).

The time lag between RAS and ISQ was determined through cross-correlation for individual waves identified in the extended tidal band (6-36 h; examples in top and bottom panels of Fig. 4). As before for the IT segmentation and F<sub>E</sub>, only clearly identifiable waves and periods with high coherence were included in the analysis. The observed time lags range from 2 up to 8 hours with lower time differences for the D band. Phase speeds were determined from the time lags and the effective distance between the stations projected on the energy flux direction (cf. vectors in Fig. 3). We assume that the horizontal direction of the energy flux coincides with the horizontal phase propagation direction, which holds for linear internal waves in a horizontally homogeneous,

weakly dissipative medium without significant background shear or refraction (Cushman-Roisin and Beckers, 2011). While both the time lags and the direction of the energy flux vary over the observed period, the estimated phase speeds exhibit an interpretable pattern.

Figure 5: Cross-shelf coherence analysis. (a) Temperature time series at 18 m depth from stations RAS (black) and ISQ (red). Bars represent  $\xi_{rms}$  for the D (grey) and SD (black) components, as shown in Fig. 3d. (b) Wavelet coherence between the two temperature series, emphasizing the tidal frequency band (D and SD periods are marked by dashed lines). (c) Phase speed based on time lags between RAS and ISQ (blue bars), derived from modal analysis (shaded grey area) and two-layer model (points). Phase averaged PEA (solid black line) is shown on the right y-axis.

On average, the phase speed from the time lag method ranges around  $0.5 \text{ m s}^{-1}$  which is slightly above the high end from modal analysis (diurnal mode-1, Fig. 5c). The  $c_p$  from SD ITs (29 Aug - 6 Sep and before 3 Oct) are only marginally lower on average compared to the D ITs. The long wavelength speed for an idealised two layer system (see Sec. 2.5) ranges slightly below the highest mode speed (Fig. 5c).

The c<sub>p</sub> we observe via the time lag method exceed the mode-1 values at certain times while during other periods the differences are small. Martini et al. (2013) and Duda et al. (2004) similarly derived speeds of ITs from signal lags between moorings over the slopes of the Oregon and South China Sea shelves, respectively. While the observed M<sub>2</sub> speeds in depth >1000m over the Oregon slope only match the lower speeds of mode-3, which Martini et al. (2013) attributed to mode scattering, the O1 phase velocities of 1.14 m s<sup>-1</sup> observed between 80 - 350 m by Duda (2004) are comparable to mode-1 c<sub>p</sub> estimates. The high phase speeds we observe appear mostly during periods of enhanced stratification as indicated by PEA. Underestimation of stratification sharpness during these periods could be a potential cause for smaller modal speeds. Other potential reason for the higher than mode-1 c<sub>p</sub> we observe are advection by background flow (Xie et al., 2015), interactions with the barotropic tide (Stephenson et al., 2016), or cross-shelf flow due to the regionally pronounced land-sea breeze in the D band.

Before 3 Oct, peaks in internal wave  $c_p$  coincide with periods of increased stratification, as indicated by higher PEA, while reduced stratification (low PEA) corresponds to lower  $c_p$ . Until the end of September, waveforms are similar in the temperature records of the two stations, and time lags are estimated by cross-correlation over entire individual wave phases. During the last recorded fortnightly IT pulse around 5 Oct, such time lag estimation is no longer possible. The rising phase of temperature (i.e., isotherm depression) appears approximately in phase between the two stations (cf. lower right panel of Fig. 4), while the cooling phase is

delayed inshore. This coincides with nonlinear steepening into bore-like structures, characterized by increased amplitude and polarity reversal of the incoming ITs. At this time, the vertical level used (18 m depth, 5 m above bottom at ISQ) is no longer suitable for tracking time lags.

Wavelengths were derived from the time lag phase speeds and the individual periods of the identified IT waves which can deviate from the fixed harmonic frequencies due to Doppler shifting. Wavelengths in the SD and D bands are concentrated between 15 to 25 km and 40 - 50 km respectively, which falls within the range of other observations on shelves for comparable latitudes (Holloway et al., 2003; Wang et al., 2022). Duda et al. (2004) consider the ratio of particle velocity to phase velocity  $u_r/c_p$  as a measure of nonlinearity. In our case the ratios range up to  $\sim$ 0.5 and while these values are below unity, they are nonetheless sufficiently large (and substantially larger than reported by Duda et al. (2004)) to be associated with nonlinear effects as discussed in previous Sections.

#### 3.4 Spectral analysis

Fig. 6 displays amplitude spectra for temperature, vertical displacement, and currents. As we have identified remote generation as likely scenario for the local ITs, we include here for comparison TPXO tides from a location at the entrance to the Strait of Hormuz (NW see Fig. 1) where IT activity has been observed (Small and Martin, 2002; Pous et al., 2004; Subeesh et al., 2025). It should be noted that remote here refers to a distance of only around 180 km away from the local shelf. Amplitude spectra (rather than power spectral densities) are used to allow direct comparison with amplitudes of astronomical tidal constituents from the TPXO model at ISQ and NW. The spectra represent the sum of FFT amplitudes from positive and negative frequencies. For the complex velocity time series (u + iv), this corresponds to the combined rotary components and is directly comparable to the major axis amplitude from tidal harmonic analysis. Clockwise motion is consistently dominant in the currents. For real-valued data, the spectrum is symmetric, and the same summation ensures consistency with constituent amplitudes from tidal harmonic analysis. The spectrum of the 22 months bottom-temperature record at the shelf-edge station in RAS resolves distinct peaks at the main tidal frequencies and a smaller peak close to the ½ M2 frequency (Fig. 6a). Both lower resolution spectra from the 3 month data at ISQ, the thermocline displacement and the baroclinic currents, follow the general patterns of the temperature with dominant peaks at K1 (and O1) and a lower peak at ½ M2 (Fig. 6 b-c).

The amplitudes of the sea level constituents from TPXO are nearly identical between the local station ISQ and the remote location at NW, both dominated by M<sub>2</sub> (Fig. 6b). The major-axis amplitudes of tidal current ellipses derived from TPXO show that local astronomical tidal currents at ISQ are strongly dominated by M<sub>2</sub>, even stronger than the sea level, with only weak energy in the D band (Fig. 6c). In contrast, the D constituents at NW are comparatively strong, though still weaker than M<sub>2</sub>. While the observed barotropic currents match with the local TPXO estimates at ISQ in the SD band, the observations are much larger in the D band. Table 1 summarizes the predominantly diurnal variability in all in-situ data whereas all data from TPXO exhibit form numbers 

Shelf in the Timor Sea. They explain the contrast as a result of seasonal stratification changes and standing wave interference that locally suppress  $M_2$  and enhance diurnal constituents.

Figure 6: Amplitude spectra for (a) bottom temperature at station RAS, (b) thermocline displacement at ISQ (red line), and sea-level amplitudes of tidal constituents from the TPXO model at ISQ (blue bars) and at NW (white bars). (c) Barotropic (black line) and baroclinic (red shaded area indicates surface-to-bottom range) currents at ISQ, with major-axis tidal current amplitudes from the TPXO model (blue and white bars for ISQ and NW, respectively).

Table 1: Form number  $K_1+O_1/M_2+S_2$  for temperature, thermocline displacement, and currents at stations RAS and ISQ from in-situ measurements, and currents and sea level from TPXO tidal predictions at ISQ and NW locations. Values greater than 1 indicate D dominance, while values less than 1 indicate SD dominance.

| Source  | Station | Variable    | $ \underbrace{ \underbrace{ \underbrace{ K_1 + O_1 } }_{ \underbrace{ \underbrace{ K_2 + S_2 } } } $ |            |
|---------|---------|-------------|------------------------------------------------------------------------------------------------------|------------|
| In-Situ | RAS     | Temperature | 1.35                                                                                                 | . <u>Q</u> |
|         | ISQ     | Thermocline | 2.04                                                                                                 | b.clinic   |
|         |         | Currents    | 3.07                                                                                                 |            |
|         |         |             | 1.95                                                                                                 | - O        |
| TPXO    | ISQ     | - Currents  | 0.20                                                                                                 | b.tropic   |
|         | NW      |             | 0.71                                                                                                 |            |
|         | ISQ NW  | sea level   | 0.63                                                                                                 |            |

While the surface tide (sea level) within the western GoO is largely homogenous, the barotropic transport has a constituent dependent structure particularly near the entrance to the Strait of Hormuz where the K1/M2 co-range lines become perpendicular and diurnal tidal transport is enhanced (Small and Martin, 2002). Both Pous et al. (2004) and Subeesh et al. (2025) report strong  $K_1$  energy and IT activity in the vicinity of the Strait of Hormuz near the NW location pointing towards a potential remote generation site for the ITs we observe on the Al-Batinah Shelf. The distinct spectral peaks near the  $\frac{1}{2}$   $M_2$  frequency that we observe, might, on the other hand, be an indication of subharmonic resonance in local  $M_2$  generation which can occur equatorward of 28.8° latitude (Gerkema, 2006).

## 3.5 Kinetic Energy

The ellipses shown in Fig. 1 were determined by least-squares fitting to the two decomposed components of baroclinic (all depths) and barotropic currents. The orientation of the major axis of the baroclinic component approximately corresponds to the general direction of the IT energy flux and is roughly orthogonal to the primarily alongshore orientation of the barotropic currents. Fig. 7a shows kinetic energy (KE) at ISQ, with the depth-averaged (barotropic) and baroclinic components derived from the tidal band signal 6–30 h as defined in Nash et al. (2012). Within the tidal band, baroclinic KE is pronounced in summer and retains approximately 10% of the summer energy during winter. Barotropic KE is larger in winter and comparable in magnitude to the baroclinic KE observed in summer. The ~20% reduction of barotropic KE in summer is substantial enough to suggest seasonal modulation of the local barotropic tides, at least in terms of current strength. Potential causes for seasonality in barotropic tides include seasonal changes in barotropic energy loss during conversion to baroclinic tides (Li et al., 2012), stratification-induced changes in barotropic tidal transport (Müller, 2012) or tidal ranges (Li et al., 2021), and nonlinear barotropic-baroclinic coupling (Duda and Rainville, 2008). This seasonality has implications for harmonic analysis of the local barotropic tide, which we briefly

The raw (total) summer KE is highly variable and, on average, significantly stronger than in winter (Fig. 7a). A longshore flow is observed in summer, which is vertically decoupled by stratification. The regional monsoon influence, as described by (DiMarco et al., 2023) and (Al-Hashmi et al., 2019), induces a persistent low-frequency westward current over the local shelf. The high variability in the summer KE results from the background flow that elevates the mainly longshore barotropic tidal oscillations away from zero, thereby distributing KE over a wider range compared to similar currents in winter oscillating around zero.

## 3.6 Predictability

address below in the context of predictability.

Predictability reflects how well IT phase and amplitude are maintained over time, and SS express this as the fraction of tidal-band variance captured by a harmonic fit, providing a basis for anticipating their effects on coastal mixing and transport. Fig. 7b displays the SS as a function of the window length in days following the method described in Nash et al. (2012). SS variations for baroclinic conditions (currents at ISQ and temperature at RAS) in summer (Apr-Oct) and winter (Dec-Feb) can be directly compared to the range from Nash et al. (2012). We also show the same analysis applied to barotropic currents at ISQ in both seasons.

SS variations of all baroclinic signals exhibit similar shapes, albeit at different absolute levels. The two summer temperature records at the shelf break show the highest SS values, exceeding the upper end of the global range at short window lengths. In contrast, the SS of the baroclinic currents fall within the mid (summer) to lower (winter) part of the global range (Fig. 7b). Owing to the similar shapes of all SS curves, the derived  $T_{50\%}$  values are comparable across signals, supporting the robustness of the estimation. We find  $T_{50\%}$  between 16 and 19 days on the Al-Batinah shelf, which lies at the upper end of the 14 global sites for which Nash et al. (2012) report  $T_{50\%}$  values.

The region with the highest IT predictability in Nash et al. (2012), comparable to our observations on the Al-Batinah shelf, is the Timor slope of the northwest Australian shelf. Rayson et al. (2021) analyzed the ITs in this area using a Seasonal Harmonic Model. Similar to our results, they observe diurnal IT dominance at most stations despite M<sub>2</sub> dominance in the local barotropic tides, and attribute IT variability to seasonal changes in stratification and the presence of both local and remote generation. On the Al-Batinah shelf, our data also suggest a significant contribution from remote generation, which contrasts with the high SS. The two regions share general geographic features: both are semi-enclosed marginal seas located at similar latitudes. These parallels, and the similarity in the observed ITs, suggest that elevated IT predictability can persist under remote forcing in certain geographic configurations when supported by stable and coherent seasonal stratification.

Figure 7: KE and IT predictability. (a) Depth averaged KW at ISQ. Qualitative histograms of the temporal variation are shown color-coded, bars represent temporal means. Shown are total (unfiltered) currents as well as band-pass filtered (6-30h) and decomposed components. Summer/winter periods are marked in Fig. 2. (b) Skill score (SS) for tidal prediction versus analysis window length. Thick lines represent barotropic flow, while markers represent baroclinic flow with power law fits (thin lines). Blue tones correspond to winter, red tones to summer. Dashed green lines indicate bottom temperature at station RAS during two summer periods 2021 and 2022. Shaded area depicts the SS range from Nash et al. (2012) at 16 global sites.

We apply the same methodology to our barotropic currents at ISQ. For stationary barotropic tides, the SS curve is expected to remain high and nearly flat, with reductions arising mainly from unresolved constituents or short-term non-tidal variability (e.g., sea breezes) not represented in the harmonic fit. The SS of the barotropic tidal fits begins around 98% for short windows but declines markedly with increasing record length (Fig. 7b). Harmonic analysis of a 2.5-month summer record yields a SS of approximately 85%, while the shorter winter record levels off somewhat higher, at around 90%.

The seasonal decline in barotropic predictability, particularly during summer, contrasts with the stationarity of unmodulated astronomical tides. This reduction likely reflects the influence of enhanced stratification, the presence of ITs, and possible observational cross-contamination in the summer records. These findings underscore the importance of accounting for IT activity and vertical structure when conducting harmonic tidal analyses of regional barotropic currents. Depth-averaged numerical models applied to this area have limited capability to reproduce realistic summer dynamics on the Al-Batinah shelf.

#### 4 Conclusions

This study presents a detailed characterization of IT dynamics on the Al-Batinah Shelf, based on an 11-week mooring dataset encompassing the late summer stratification regime and its autumnal breakdown. The ITs observed are predominantly diurnal, despite semidiurnal dominance in the barotropic forcing, indicating modulation by regional stratification and potential remote generation, possibly linked to the Strait of Hormuz.

The mean thermocline depth, buoyancy frequency, and PEA together characterize a highly stratified shelf environment that supports the shoreward propagation of coherent ITs. Energy fluxes and waveform structure indicate a transition from quasi-linear, depression-type waves to increasingly nonlinear waveforms, including steepening, skewness, and eventual polarity reversal—features consistent with bore-like evolution. Vertical modal analysis further confirms a shift from mixed-mode to first-mode dominance as the thermocline deepens and stratification weakens.

Cross-shelf coherence and phase speed estimates demonstrate that ITs maintain coherence into the inner shelf, with observed phase speeds occasionally exceeding linear modal predictions. This discrepancy is attributed to nonlinear effects, stratification variability, and possible background advection. Spectral analysis confirms the diurnal dominance in IT variability and reveals a subharmonic peak near  $\frac{1}{2}$   $M_2$ , suggestive of local nonlinear interactions.

The kinetic energy and predictability analyses show seasonally modulated baroclinic activity with skill scores and T<sub>50</sub> values comparable to the higher end of global comparisons. Summer barotropic currents show reduced predictability, potentially due to contamination by ITs and stratification-induced variability.

Together, these findings offer a coherent picture of IT evolution in a stratified marginal shelf setting and highlight the importance of nonlinear dynamics and remote forcing. Future work incorporating spatial observations and modeling will be necessary to fully resolve the propagation pathways and energy dissipation zones of the observed ITs.

- Data availability. The mooring and CTD datasets analyzed in this study will be deposited in a public repository with DOI before final publication. Until then, the data are available from the corresponding author upon request. In addition, publicly available data products were used: OSTIA satellite sea surface temperature dataset (Good et al., 2020; https://doi.org/10.1029/2019JD032423). TPXO barotropic tide model (Egbert and Erofeeva, 2002; https://doi.org/10.1029/2001JC001210).
- *Author contributions*. GAB designed the study, carried out the field measurements, performed the data analysis, and wrote the original draft of the manuscript. EF and BYQ contributed to the investigation and to writing review and editing. RAH contributed to methodology development and to writing review and editing.

Competing interests. The authors declare that they have no conflict of interest.

*Acknowledgements*. We thank Sultan Qaboos University staff Badar Al-Buwiqi, Farid Al-Abdali, Salim Al-Khusaibi and Antoine Leduc for their support during field operations.

Financial support. This work was supported by ONR-GLOBAL grant no. N62909-21-1-2008 and SQU grants RC/RG-570 AGR/FISH/20/01 and RF/AGR/FISH/24/01.

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
