# Peer review of "Internal tides on the Al-Batinah shelf: evolution, structure and predictability"

_EGUsphere, 2025_

## Author Comment (AC1)

RC1: 'Comment on egusphere-2025-4158', Johannes Becherer, 02 Oct 2025

The manuscript presents data from two moorings on the Al-Batinah shelf and provides a detailed analysis of the internal tide at this site, covering most commonly used characteristics. This work is a valuable contribution to the literature on continental shelf internal tides, adding a new data point to the global record. Notable findings include a predominantly diurnal internal tide despite stronger semidiurnal forcing, a subharmonic peak at 1/2 M2, and a high predictability score despite likely remote generation. The manuscript is well written, and the figures are of excellent quality. I recommend publication after minor revisions.

We thank Dr. Becherer for the thorough review and constructive feedback. The insightful suggestions were very helpful in improving both the content and presentation of the paper. Our replies to the reviewer's questions are listed in red after each reviewer's comment.

Specific comments:

L32: (~20km) is not really wide compared to other shelf regions. I would still call that a rather narrow shelf compared for instance to the US west coast or the NW European shelf.

In the context of L32 we meant "wide" relative to the other shelves within the Gulf of Oman. We have clarified the description.

L48: steep -> sharp?

Yes, in L38 we mean a "sharp" thermocline and have changed the text. The original "steep" referred to the vertical thermal gradient.

L65: I don't get it. What is the 285-115° axis?

The mentioned 115° was a typo. It should be 285-105° - roughly along the WNW-ESE axis. We have updated the text.

L83: 15? Resolutions

Thank you, units were missing, we have updated the text to: "15 min temporal resolution".

L90f: I am confused by this zTC calculations. You say you calculate it not along a fixed isotherm but as a fixed DT to SST. SST comes from satellite, right? So how consistent is that. I was also curious how a diurnal change in SST would affect zTC, but I guess the satellite data is daily averaged? If the diurnal SST signal is included this could introduce an artificial diurnal signal in zTC. Please clarify.

In our case surface temperature refers to the uppermost temperature from the mooring, which is  $\sim$ 3 m below the actual surface (Fig.3 b) but practically always within the upper (well) mixed layer. We have added the description to the text.

RD: Also, Is there a diurnal sea breeze that could influence the diurnal signal?

Yes, in our region there is a diurnal sea breeze and we did look into possible connections to the ITs. We actually did this in depth, assessing sea breeze as a potential IT driving mechanism, but could not identify clear interactions.  $S_1$  is close to  $K_1$  so a definite spectral separation for those frequencies is not possible from our data but clear baroclinic peaks in  $O_1$  and  $M_2$  do not correspond to wind. Also, periods of increased diurnal IT activity (fortnightly) do not coincide with increased sea-breeze periods.

L91 ztc -> zTC

Thank you, we have updated the text.

L106: So if your diurnal band includes the inertial frequency, how can you be sure that your diurnal tide calculations are not contaminated by inertial motions?

We mention that f is included in our frequency window more for completeness. The higher resolution spectrum of the 22-month temperature record at RAS does not show a significant peak at the inertial frequency. Therefore we do not consider inertial contamination to be significantly affecting our analysis. In order to retain instationarity we did not use stationary band pass filters (which might allow sharper band definition and thus the exclusion of f, albeit being relatively close to  $O_1$  (and very close to  $O_1$ ). Instead we used wavelet decomposition/reconstruction which has fixed band widths relative to the input data.

eq5: You don't introduce Psi

Thank you, we have updated the text.

L255: I cannot see stratification breaking down in Fig.3b

Correct, our formulation was not precise. What we mean to say is that stratification eventually starts to decline towards winter conditions when the shelf sea is fully mixed. In our ISQ 2022 record we do not fully reach this point yet. We have corrected the formulations in the text.

L259: Where does this S2 estimate come from. It looks like internal wave shear can be much larger at times. Are you sure that Ri > 1/4 always? This seems a bit handwayy.

We calculate  $S^2$  from the ADCP data. In the text we mention "mostly Ri >  $^{1}/_{4}$ ", and "predominantly stable". Ri across the thermocline is >  $^{1}/_{4}$  for ~90% of the record-time. We have updated the text to improve precision.

L264 How can you be so sure they have not yet reached saturation?

In the context of L264 we refer to the ratio between IT amplitude and local waterdepth. We do not observe IT amplitudes approaching the total water depth at ISQ. We deduce from (Becherer et al. 2021 b) that if the IT amplitude has not yet reached a value comparable to the local water depth, then the internal tide has not yet reached saturation. As we have added a more in-depth

assessment of actual saturation as suggested (see discussion on the subject below), we have removed the explicit reference to "saturation" here.

**Fig 3:**

why is PEA jumping around so much?

PEA in original resolution (15 min) considerably varies within the internal tidal phase. The daily PEA as shown in Fig.3b was determined by averaging in strict 24h windows which shift across IT phases and thus introduce variability. We have now used 36h overlapping windows and the PEA "jumpiness" is reduced. The remaining variability on few-days timescales is mostly linked to wind, which we have assessed but do not present to avoid overloading the already long paper.

panel c: I don't understand the triangles. In general I feel that such a complex figure requires a more detailed caption.

Triangles represent troughs (pointing down, size scaled by the ratio trough/crest) and crests (pointing up, size scaled by the ratio crest/trough) of the diurnal band-passed thermocline. So the two triangles of one IT indicate polarity and its magnitude. We have updated the caption.

panel d: I don't understand what the background lines represent? Are they described in the caption.

Background lines represent amplitude modulation of the TPXO derived barotropic transport (converted to vertical velocities (new)) at the local shelf slope for the D and SD bands. We have updated the caption. We show this to visualize the large phase lags of several days between local barotropic and baroclinic tides.

The arrows indicating energy flux are nice, but they make it hard to gauge the magnitude. I wonder if you should add a panel with the fluxes magnitude as a time series.

The new panel (d) in Fig. 3 shows the timeseries of the total (low pass filtered) incoming energy flux  $F_{E}^{in}$  (see also discussion below).

L281: I wonder if the delay in the fortnightly cycle could help you with the remote generation argument. If you can pinpoint the delay and combine it with the phase speed of the internal tide you might be able to estimate a distance to the generation site.

Yes, this aligns with our line of reasoning. We had originally planned to integrate the suggested analysis into this paper, but found that it would over-extend the length of one paper. We study generation and propagation of the regional ITs in detail in a follow-up paper which will be submitted shortly.

L285: This statement is not quite right. While the flux we observed in California dropped by at least an order of magnitude at 25m, it is still at a comparable level to the fluxes you observe O(1-3) W/m. This meant full saturation in our case, because our stratification was weaker than yours.

Thank you for pointing this out. We have updated the discussion to more accurately reflect the cited literature. Also please note that the magnitude-scale of  $F_E$  that was originally shown in Figs.1 and 3 was mislabeled. The values of 1 Wm-1 was actually the magnitude of the depth averaged Flux (Wm-2). We have corrected the scale vectors to correctly represent the depth integrated magnitude (Wm-1).  $F_E^{in}$  ranges between 5 and up to ~30 Wm-1. The new panel (d) in Fig. 3 shows the magnitude of the total 36h low pass filtered depth-integrated onshore-directed energy flux which is also discussed it in the added text passages.

On that note: Since you think that you are still outside of the saturation regime, it would be interesting to estimate where the saturation regime would start in your case. You could use (eq 15, Becherer et. al.) to calculate this. From my rough back of the envelope calculation I get around 12-15m depth, which is not too far away from your shallow mooring.

You could also check if you are really outside of the saturation regime by estimating the saturated flux based on your observed top to bottom density difference (eq 10, Becherer et. al.) and compare it to your observed fluxes.

Thank you for the valuable suggestion. We have estimated saturation conditions for our ISQ data following the suggestion. We have added a panel to Fig. 3 and included a new paragraph into the manuscript and updated passages where saturation is discussed.

Another way to check would be to compare the IT signal observed at RAS with ISQ and see if you can detect already some amplitude decrease. You sort of do this in Fig5a, where it looks like that could be a decay at least in the early part of the record. So I am not 100% convinced yet that you are outside of the saturation regime.

We agree that comparing displacement amplitudes would be informative. But we have only accurate displacement for ISQ; for RAS we do not have a full-depth mooring. Comparing amplitudes of temperature fluctuations at a fixed vertical level (data in Fig 5) could be done but we think that uncertainties would be to large for a confident assessment. Generally the thermocline is sharp and its local vertical level is modulated by low frequency cross-shelf MLD variations. IT signatures at fixed-depth temperature records are quantitatively sensitive to this.

L296: What do you mean by IT intensity here? This causal connection between IT intensity (??) and mixing implies that mixing is happening close to your site, which also implies that the IT is already substantially dissipating, which in turn implies that it reached saturation?

By IT intensity we mean  $F_E$  or displacement-RMS. In this context we referred primarily to the fortnightly patterns not immediate PEA reduction during large individual ITs. At fortnightly frequency, PEA reduction after increased  $F_E^{in}$  is observable, so we thought of it more as a

"feedback from downstream" mechanism indicating the shifting of the saturation zone during the fortnightly cycle. We now place this into the context of the extended assessment of the saturation regime.

L299: But how far onshore? See comment above for saturation regime.

In the additional analysis we estimate Hs to range between 15-20 m in summer which corresponds to a distance of 4 to 8 km from the shore. In the autumn transition Hs reaches ISQ (~10 km offshore) at the beginning of October. We have updated this in the new passages in the manuscript.

L300: You should also cite McSweeney et al. 2020 here (reference below). This reference could also be interesting in the context of the phase speed estimates and the polarity reversal.

Thank you for the suggested paper. We have integrated it at several points to broaden the discussion context.

L303: What is your temporal resolution. Are you saying that you cannot resolve the buoyancy period?

The temporal resolution of our data is 15 minutes (Nyquist 30 min), so yes, we do not resolve the local buoyancy period ( $T_b \sim 3-15$  min) and therefore cannot fully assess ISW/NLIW whose spectral energy can extend down to  $T_b$ . We mentioned in the text that our data does not resolve high frequencies and we therefore cannot properly assess ISW/NLIW. We have now removed the sentence as it does not add substantially to the paper.

Fig4: - You need a colormap here. Otherwise this figure cannot be used as stand alone.

Thank you. We have added a colormap.

L350: "were derived from zero-crossing  $N^2(z)$  profiles" How can the N2 profile have zero crossings? Do you mean density profile here?

What we mean here is that we use  $N^2$  profiles at those times when the instantaneous TC crosses the low-passed TC. We do this to retain a sharp  $N^2$  peak, as opposed to using low-passed (or phase averaged)  $N^2$  profiles which would vertically smear the  $N^2$  peak. The method is mentioned in sec. 2.3 and we have updated the text to improve clarity.

L381: I don't follow this argument. How does the coherence confirms the DWL do not influence IT?

We agree that our original formulation was not conclusive. We mislabeled diurnal full-depth warming that we sometimes observe in the shallow inner shelf as DWL. Actual DWL can be deeper offshore. Our reasoning was that this effect extends from the inner shelf to a certain distance offshore towards ISQ and that the high coherence between ISQ and RAS indicates that

the effect has not reached ISQ. As we have too little data to substantiate this, we have removed the statement. Thank you for pointing this out.

L406: This is again a good place for the McSweeney reference.

Agreed, we have added it with a brief discussion.

L412: The advection by background currents should be testable with your ADCP data.

We know from other shelf mooring data that, contrary to the ITs, background flow can be rather variable across the local shelf (between shelf-edge and inner shelf). We have therefore refrained from assessing potential advection from the ISQ data alone.

L423: This suggests that the distance of the two moorings is smaller than the wavelength, which probably also explains the large coherence.

Correct, the mooring distance is smaller than the wavelength of the diurnal ITs. We have added the qualification in the text.

Section 3.4: It is not clear why you compare the NW position here. Do you have good reasons to think that this could be the generation site for the observed ITs? If yes please explain. You could also test if the distance matches the delay in the fortnightly cycle you observe, which would be a nice consistency check.

As mentioned above, we did a parallel study of potential IT generation in the Gulf of Oman in which we identified increased energy conversion in the area around the NW location. As the other study has not yet been published, here we point to the literature that describes IT observations in that region.

**Fig6:**

b) I am really surprised that the remote and the local station lie practically on top of each other in terms of the barotropic tide (blue and white bars). Is this correct?

We looked into the regional TPXO data carefully. The sea level amplitudes are indeed very close (not identical) between the remote and local locations. Within the GoO there is no distinct "amphidromic structure" unlike in the adjacent Persian Gulf.

c) Is this just by chance that the barotropic and baroclinic tide have almost identical amplitudes?

Fig. 6 is for summer conditions. In winter, barotropic currents are still similar, while IT signals are strongly reduced. Cross contamination between the two (BC / BT) can however not be entirely ruled out as we discuss e.g. in L531. To answer the question, we have not found a specific dynamic that would explain the similarity in summer amplitudes.

**Fig7:**

a) you don't mention what the two bars for each band represent. I assume its summer and winter, but this is not stated. Also there is no colobar.

Yes, the red and blue bars represent summer and winter. All legend labels are unified in panel b. We have added a colorbar for the histogram.

caption KW -> KE?

Thank you. Corrected.

---

## Author Comment (AC2)

RC2: 'Comment on egusphere-2025-4158', Anonymous Referee #2, 16 Oct 2025

This paper presents a detailed analysis of measurements on a shelf in the Gulf of Oman that show predominantly diurnal internal tides. The results are carefully described and offer a valuable contribution to the understanding of internal tides in that region.

We appreciate the reviewer's thoughtful feedback and valuable suggestions, which have strengthened the manuscript. Our replies are listed in red after each reviewer's comment.

I recommend publication after a minor revision, in which the following points are considered.

The authors suggest that the diurnal internal tides may be generated remotely because locally the barotropic tidal current is mainly semidiurnal. This could be correct, but it would be worthwhile to consider the local setting more fully, perhaps with a simple internal-tide generation model. Diurnal and semidiurnal internal tidal beams have very different slopes and the local bathymetry may be more favorable to generate one or the other.

We agree that from the data presented in this manuscript alone, local generation can not be definitely ruled out. We have added qualifications in the respective sections (e.g. L265). Local shelf slope topography could indeed favor energy conversion in the diurnal band. We have in a parallel study performed a regional assessment of energy conversion, slope criticality and regional propagation. Since the present manuscript is however already long (as commented below) these results will be presented in a separate manuscript which is in preparation.

Another argument put forward in the paper is the shift and variability in fortnightly cycles between barotropic and baroclinic signals. However, such shifts have been demonstrated to occur even in locally generated internal tides, because the M2 and S2 (or similarly K1 and O1) beams propagate at different angles, creating spatially varying phase shifts in the baroclinic fortnightly cycles, plus a sensitivity to time-varying background stratification.

We agree that interference both vertical, from different beam angles, and horizontal, from separated generation locations, can produce spatially variable phase shifts in baroclinic fortnightly cycles. We have added qualifications in the respective sections (e.g. L256) with reference to relevant literature (Gerkema 2002).

As a general point, I think the paper could be shortened (leaving out details that do not really add much to the story), highlighting the main findings. It is now quite a long paper on a relatively limited dataset.

We have revised the whole manuscript and have shortened sections where possible.

Besides, I notice a number of smaller points that deserve attention in a revision:

1) line 88, if rho denotes (in-situ) density then the expression for N2 should have a term involving the speed of sound as well.

Thank you, we used potential density and have now specified it in the text.

2) eq. (7) is based on the hydrostatic approximation (which is fine for internal tides), not "non-hydrostatic" as stated in line 167.

Thank you for pointing this out. We have corrected the text. The reference to "non-hydrostatic" was a leftover from an earlier draft in which we considered solving the full non-hydrostatic vertical-structure equation. Because the non-hydrostatic corrections remain negligible during periods of weakened stratification, we reverted to the hydrostatic formulation.

3) In figure 3d, what direction do the vectors refer to? (cross-slope?)

Vectors are in geographic orientation with north up (north arrow is shown in the figure).

4) In line 422, the authors state, as if it were evident, that internal tidal frequencies can deviate by "Doppler shifting". How? In any situation where both the source and observer are fixed in space (as is the situation here), no Doppler shifts occur (and the presence of a mean flow does not change this!). In other words, this statement needs a more careful consideration (or perhaps removal).

Our point concerns the Eulerian phase modulation of the internal tide at a fixed site due to space- and time-varying background currents that advect and refract the wavefield. In that case, the observed phase rate can differ from the intrinsic value ( $\omega_{obs}$ = $\omega$ +k · U), producing apparent shifts within the diurnal band. We thank the reviewer for correctly pointing out that this should not be called Doppler-shift. We have replaced "Doppler shifting" with "phase modulation by space- and time-varying background currents".

---

## Author Response (AR2)

**Referee Comment:**
The authors have addressed most of my concerns through this revision, except for the last point. It is not merely a semantic issue whether one should call this Doppler shift or not, but also (and more importantly) a question that really pertains to the physics. For internal tides, generated over topography (a feature fixed in space, i.e. the source is not moving) and observed at moorings (equally fixed) there would be no change in frequency. The authors mention the "intrinsic frequency" but this is the frequency measured by an observer moving with the mean flow - which has no bearing on the situation described in this paper. (An elementary exposition on this topic can be found in Gerkema et al., J. Phys. Oceanogr. 43, 432-441, 2013)

**Authors response:**
We were referring to potential effects of *space- and time-varying* background currents. Since we did not actually analyse this aspect and it is not essential to our study, we have removed the passage from the manuscript.

**Revised manuscript passage:**
Wavelengths were derived from the time-lag phase speeds and the individual periods of the identified IT waves.